# The Implementation of Childhood Obesity Related Policy Interventions in Malaysia—A Non Communicable Diseases Scorecard Project

**DOI:** 10.3390/ijerph18115950

**Published:** 2021-06-01

**Authors:** Wilfred Kok Hoe Mok, Noran Naqiah Hairi, Caryn Mei Hsien Chan, Feisul Idzwan Mustapha, Thamil Arasu Saminathan, Wah Yun Low

**Affiliations:** 1South East Asia Community Observatory (SEACO), Monash University Malaysia, Subang Jaya 47500, Malaysia; wilfredmok89@gmail.com; 2Global Public Health, Jeffrey Cheah School of Medicine and Health Sciences, Monash University Malaysia, Subang Jaya 47500, Malaysia; 3Centre for Epidemiology and Evidence-Based Practice, Department of Social and Preventive Medicine, Faculty of Medicine, University of Malaya, Kuala Lumpur 50603, Malaysia; noran@um.edu.my; 4Faculty of Health Sciences, Universiti Kebangsaan Malaysia, Kuala Lumpur 43600, Malaysia; caryn@ukm.edu.my; 5Disease Control Division, Ministry of Health Malaysia, Putrajaya 62590, Malaysia; feisulidzwan@gmail.com; 6Institute for Public Health, National Institutes of Health, Ministry of Health Malaysia, Shah Alam 40170, Malaysia; thamilarasu.s@moh.gov.my; 7Asia-Europe Institute, University of Malaya, Kuala Lumpur 50603, Malaysia

**Keywords:** childhood obesity, Malaysia, policy interventions

## Abstract

(1) *Background:* The prevalence of overweight and obesity among children has increased tremendously in the ASEAN region, including Malaysia. In Malaysia, the National Strategic Plan for Non-Communicable Diseases (2015–2025) provides the overall framework for its response to the non-communicable diseases (NCD) epidemic. Preventing childhood obesity is one of the key strategies for early intervention to prevent NCDs. The objective of this research is to examine the current status of policy interventions in addressing childhood obesity in Malaysia. (2) *Methods:* A panel of 22 stakeholders and experts from Malaysia, representing the government, industry, academia and non-governmental organizations, were sampled using a modified Delphi technique. Data were collected using a modified NCD scorecard under four domains (governance, risk factors, surveillance and research and health systems response). A heat map was used to measure the success of the four realms of the NCD scorecard. For each domain of the NCD scorecard, the final score was grouped in quintiles. (3) *Results:* A total of 22 participants responded, comprising of eight (36.4%) males and 14 (63.4%) females. All the domains measured in implementing policies related to childhood obesity were of low progress. Nine governance indicators were reported as 22.5% (low progress), four in the risk factors domain, and two in the surveillance. This shows that timely and accurate monitoring, participatory review and evaluation, and effective remedies are necessary for a country’s surveillance system. (4) *Conclusion:* Although Malaysia has published several key strategic documents relating to childhood obesity and implemented numerous policy interventions, we have identified several gaps that must be addressed to leverage the whole-of-government and whole-of-society approach in addressing childhood obesity in the country.

## 1. Introduction

Non-communicable diseases (NCDs) have become the highest priority target for public health in high, medium and low-income countries, respectively [1]. Most risk factors and elevated risk behaviors upstream of NCDs have no signs, and people also do not associate day-to-day behavioral decisions with subsequent illnesses [2,3]. The rate of obesity has doubled in more than 70 countries since 1980, as analyzed in a comprehensive review of the Global Burden of Disease (GBD) Research. In total, 107.7 million children and 603.7 million adults were obese in 2015 [4]. In 2015, the prevalence of obesity among children worldwide was only 5 per cent and overweight and obesity combined is 23 per cent [4,5].

In Malaysia, the prevalence of childhood obesity (children under 18 years) increased from 6.1 to 11.9 per cent between 2011 and 2015 and hit an all-time high of 14.8 per cent in 2019, according to the National Health and Morbidity Survey (NHMS) [6,7,8]. The South-East Asian Nutrition Study (SEANUTS) revealed 14.4% overweight and 20.1% obese among urban children aged 7 to 12 years of age in six regions in Malaysia [9]. Alarmingly, the rise in childhood obesity in many countries has surpassed adult obesity [4].

Malaysia is a part of the Southeast Asian region and is recognized as Asia’s rice bowl. As a result, Malaysians eat a variety of carbohydrate-based foods [10]. Excess carbohydrate consumption, on the other hand, can lead to health issues such as obesity [11]. Obesity is linked to the infiltration of monocytes into adipose tissue, which results in inflammation and metabolic disorders [12]. Blueberry and barley were found to have anti-diabetes, anti-obesity, anti-cancer, antioxidants, and anti-inflammation properties [13,14]. These findings suggested that barley and blueberry are beneficial to a balanced diet and the development of early human intelligence.

The World Health Organization [15] has established tracking and monitoring for obesity prevention and NCDs as key tasks for researchers as civil society members. A global strategy for managing and preventing NCDs and a reporting mechanism to assess progress on 25 indicators against nine goals were approved by the 2013 World Health Assembly [15]. The National Strategic Plan provides a framework for all key stakeholders in Malaysia to mitigate the potentially avoidable and preventable burden of morbidity, death and injury related to NCDs by multi-sectoral coordination and cooperation at the national and national level. The common behaviorally modifiable risk factors for NCDs are tobacco use, harmful alcohol use, unhealthy diet such as the consumption of fewer fruits and vegetables, high salt and trans-fat consumption, and physical inactivity, while overweight and obesity, elevated blood pressure, increased blood glucose and abnormal blood lipids are the common metabolic risk factors [16]. Therefore, preventing childhood obesity is one of the key strategies for early intervention to prevent NCDs [17]. 

Measuring and evaluating progress and facilitating action by countries in reducing the burden of NCDs is essential and important, and a range of stakeholders utilize methods to monitor progress [10,11,12]. However, only a few studies have published stakeholder viewpoints on Malaysia’s development and follow-up action on NCDs [18]. Hence, this study aimed to describe the NCD policy interventions and measure performance across three areas—governance, risk factors and surveillance/research addressing childhood obesity in Malaysia.

## 2. Methods

Ethics approval for this study was obtained from the National Medical Research Register (Ref: NMRR-20-1859-55266 (IIR) and University Malaya Research Ethics Committee (Ref: UM-TNC2/UMREC-890). A written informed consent form was also obtained before participation in the study. We applied the Delphi technique as it does not demand proximity or a face-to-face meeting and thus allows for experts’ independent thought [19]; there is less opportunity to (be forced to) conform with the dominant view [20]. The Delphi method was adopted in this study, which is based on a controlled indirect association among experts (participants with knowledge of the obesity and policy options), with the experts’ judgments attempting to converge as the experiment proceeds. Participants in the group are recognized and confirmed as domain authorities [21], while researchers continue to seek a wide variety of expert viewpoints on those parameters. A minimum number of 10 panels are the most frequent sample size [22].

At first, Delphi communication consisted of face-to-face meetings between a researcher and the expert being interviewed, but this activity quickly obtained an additional mode: communication through online interactive network channels, also known as e-Delphi [23]. From the outset, no face-to-face communication among experts has been part of Delphi. Experts are, in fact, anonymous when it comes to one another.

### 2.1. Study Design

This was a cross-sectional study. This approach was used to infer potential interactions or to collect preliminary evidence on obesity policy options to help further study and experimentation.

### 2.2. Target Respondents

Purposive sampling was conducted for this study. We distributed the scorecard to the 22 participants who constituted important stakeholders and experts in this area. This was further supplemented with face-to-face interviews if the participant wanted to express further views. Respondents’ selection for this study was similar to previous studies: “A scorecard for tracking actions to reduce the burden of non-communicable diseases” published by Roman [24]. There were 22 participants that were shortlisted from each of various stakeholders from areas of government officials in health, finance, and education ministries; leaders of non-governmental organizations; private sector leaders, and academics. Inclusion criteria: Top management officials in relevant organizations mentioned earlier (e.g., director, assistant director, chairman, vice-chairman). Exclusion criteria: Officials who were retired and a past member of any organizations. A list of potential respondents was shortlisted by research team members. The principal investigator finalized the respondents from a list of identified stakeholders after obtaining a consensus from research team members.

### 2.3. Respondents Recruitment

An invitation was sent via e-mail to each potential expert asking for her/his participation. A total of a maximum two-week period was given to obtain their response. Reminders were sent to those who had not responded by day 4, 7 and 10 from the first email. The respondent had the right to refuse or withdraw from the study at any point of time during the survey. Respondent refusal or withdrawal from the survey did not affect them in any way. Once written informed consent had been received by email, the survey form was given via e-mail to the respondents. Respondents were asked to complete the questionnaire within four weeks from their consent date. Please refer to the data collection flow chart (Figure 1) for more details.

### 2.4. Data Collection

The data collection was conducted from 1 July 2020 to 30 September 2020. Data collection was carried out via email. Each respondent was sent an email inviting them to participate in the study with the participant information sheet attached (PIS) and consent form. Data collection was performed by all research team members. In addition to questionnaire distribution, the research team members also searched for relevant national statistics to complement and validate respondents’ data/ feedback.

### 2.5. Survey Instrument

Table 1 shows how each domain assessed the percentage of success of the NCD scorecard. The NCD scorecard was used to collect data. The NCD scorecard was developed from the United Nations political declaration after the high-level meeting on the Prevention and Control of Non-Communicable Diseases in 2011, following extensive consultation with global experts by a team from GRAND South—a network of 11 centers conducting research, building capacity, and advising on policy in relation to NCD in low- and middle-income countries. The score card included indicators in 3 domains: governance, risk factors, and surveillance and research. Indicators of progress were scored by stakeholders from 0 (no activity), 1 (present but not adequate), and 2 (adequate) to 3 (highly adequate) and then the percentage of progress in each domain was calculated, representing the current situation in each country. An overall score was calculated for each indicator using the median. The sum of the median scores for each domain were divided by the total possible score and presented as percentage for the country.

## 3. Results

A total of 22 experts responded to this study out of 45 experts who were approached, comprising eight (36.4%) males and 14 (63.4%) females. The mean age was 47 years old (SD: 9.66). These 22 participants included six key persons from the Ministry of Health, Ministry of Education, Ministry of Sports and Youth; eight academic nutritionists from public health nutrition researchers or universities; six stakeholders from organizations or institutions that are actively involved in combating obesity in Malaysia, and last but not least, two from industries that promote healthy eating and physical activity.

Table 2 illustrates the percentage of progress for each indicator. There are nine indicators under the governance domain, four indicators under risk factors, two indicators under surveillance and system, showing 22.5%, 22.6%, and 24.4% of progress.

Table 3 shows the policies, programs, or plans available in Malaysia in these various domains. The three main domains of the NCDs scorecard, such as the governance, risk factors and the surveillance and research components, were further analyzed based on the number of stakeholders who responded. The lowest response rate was 81% (*n* = 17) for the domain of risk factors, specifically of the indicator on a “policy, strategy, or action plan to increase consumption of fruit and vegetables that were implemented and regularly monitored.” This result reflects the uncertainty of the stakeholders on the National Agro-Food Policy, 2011–2020.

Several stakeholders mentioned in the scorecard that the current policy and action plans were not widely communicated or disseminated to the public due to the lack of any observable activity carried out under the strategic plan, such as the National Strategic Plan for Active Living (NASPAL) 2016–2025. The stakeholders also reiterated that they did not believe that the identified strategies were systematically or extensively implemented. In fact, there is a mention of NASPAL in the mid-term review of the 12th Malaysia Plan.

Regarding the existence of units within the Ministry of Health, namely, the Disease Control Division, Health Education Division, Family Health Development Division and Nutrition Division, the stakeholders expressed confusion that the general public did not have a greater awareness of the type of services provided. Regarding the existence of a unit or branch or department in the Ministry of Health that is well-resourced, more efforts were seen on communicable diseases than NCDs by the Disease Control Division.

## 4. Discussion

Our findings indicated that all of the domains measured in implementing policies related to childhood obesity were of low progress. Of particular concern were several key indicators in the governance, risk factors, research and surveillance domains. Indicators should be measurable and sensitive to changes in policy and practice. Our findings indicate that timely and accurate monitoring, participatory review and evaluation, and effective remedies are necessary for a country’s surveillance system. The prevalence of NCD risk factors remains unknown to affected individuals, consistent with the documented literature [3]. A high proportion of the population was not aware of NCD risk factors in population-based surveys. This is in line with the findings from the NHMS 2019, whereby one in three adults have low health literacy, which defines the ability to find, understand, and use health information and services needed for everyday health decision-making. This may explain why NCD risk factors remain unknown to the population in Malaysia.

The 2010–2014 National Strategic Plan for Non-Communicable Diseases (NSP-NCD) was first released in December 2010 to improve Malaysia’s response to the growing burden of NCDs, (MOH 2010) and this was followed by the National Strategic Plan for Non-Communicable Diseases (NSP-NCD) 2016–2025, published in 2016 [17]. The low progress of the domains of governance, risk factors, and the research and surveillance in Malaysia’s context to combat childhood obesity certainly indicates that many more strategies are needed to strengthen the implementation and review the policies’ effectiveness. NSP-NCD 2016–2025 has integrated the WHO NCD global monitoring goals. It underlines the vital value of an efficient surveillance framework for measuring, monitoring and understanding patterns in risk factors, morbidity and mortality, and public health response policies and interventions.

### 4.1. Governance

In terms of governance, the inclusion of NCDs in the national health plan and national development agenda, Malaysia has two relevant documents concerning childhood obesity: the Policy Options to Combat Obesity in Malaysia and the National Strategic Plan for Active Living 2016–2025. Policy Options to Combat Obesity is not a strategic policy document. It compiles policy options for the policymakers through what was a prioritization exercise. There is always a mention of NCDs in any national development agenda. Even in the 12th Malaysia Plan proposed, NCD prevention strategies were incorporated into the social re-engineering strategies. However, these mentions are seldom accompanied by its designated budget or an appropriate follow-through plan for agencies outside of the health sector. While policy options are deliberated at length and prioritized, not all stakeholders believed they were implemented to any significant extent, if at all.

Regarding the responses of some stakeholders who stated that the current policy and action plans were not widely disseminated to the public, there is a unit in every single division responsible for NCDs. For example, there is a unit specifically dedicated to cardiovascular disease prevention and another unit on cancer prevention in the Disease Control Division. In addition, the Nutrition Division is dedicated to nutrition and physical activity issues. Unfortunately, those units are not well-equipped in terms of human resources and budget allocations commensurate with the burden. This may explain why some of the stakeholders did not realize the implementation of the current policy, such as NASPAL 2016–2025 in the country.

In Malaysia, there are national projects, programs and initiatives in the health sector to mobilize resources, build capacity, train health workers, and exchange information, lessons learnt and best practices on reducing the burden of NCD. The School Health Program and Adolescent Health Program includes a mental health component and the Prioritizing Food Policy Options to reduce obesity in Malaysia 2015, of which stakeholders did not appear fully aware. In these programs, body mass index (BMI) measurement for school children is carried out by the school teachers and recorded in the students’ health record books. However, no action is taken if the BMI of the children is overweight or obese. This issue should be taken into consideration and action by the Ministry of Health to maximize and fully utilize the BMI data taken at the school. There is the inadequate dissemination and empowerment of the community in the areas of nutrition and physical activity. Overall, nutrition promotion to the population needs to be systematically implemented at all times, throughout the country, by using various channels and engagement with multi-sectoral of stakeholders. For instance, there is limited implementation of activities under the National Plan of Action Nutrition, currently monitored under the Nutrition Division’s aegis, Ministry of Health.

### 4.2. Risk Factors

The National Plan of Action for Nutrition of Malaysia (NPANM) III 2016–2025, which refers to the nutrition programs and activities for school children, is the policy strategy or action plan to reduce sugar beverage consumption among school-age children and adolescents, with regular monitoring. While this policy exists, the implementation and enforcement leave much to be desired. The poorly enforced School Canteen Guideline (under the Nutrition Division, MOH) and Guidelines on the Prohibition of Sales of Foods Outside of School Perimeters (policy under NCD section, enforcement by local authorities) are some of the examples that need to be reviewed.

We received feedback that adequate vending machines should be filled with mineral water rather than sweetened canned drinks. The price of drinking or mineral water sold needs to be capped. Schools should be provided with sufficient funding to set up water dispensers. The focus should not be on just the consumption of high sugar beverages. NPANM has outlined various programs and activities for promoting healthy eating and active living among school children. However, even from NPANM I (1996), these were not implemented and need to be systematically incorporated. This can be key to the prevention of malnutrition among children, with school children being an important target. The nutrition of future Malaysians lies in empowering the children of today. Therefore, nutritionists should be assigned to schools to implement the various activities, as there are currently none serving in this capacity in the entire Ministry of Education, Malaysia.

Next, the National Plan of Action for Nutrition of Malaysia (NPANM) III 2016–2025 is the policy, strategy or action plan to increase the consumption of fruit and vegetables implemented and monitored. This seems to show minimal progress from NPANM I, 1996, as our National Health and Morbidity Survey (NHMS) repeatedly demonstrated the low consumption of fruits and vegetables. There is a whole section dedicated to this in the NPANM, with regular monitoring through NHMS. Perhaps, collaboration with other stakeholders such as the Ministry of Agriculture (MOA) is desirable.

Regarding the National Agro-Food Policy, 2011–2020, of which most of the stakeholders are uncertain or unaware, the government could subsidize local fruits and vegetables to increase fruits and vegetables’ consumption. For example, currently imported fruits such as oranges and apples are cheaper than papayas, pineapples and guava. In terms of fruits and vegetable intake, 95% of Malaysian adults did not consume the recommended daily amount of fruits and vegetables [8]. From the observation of stakeholders about the National Coordinating Committee on Food and Nutrition (NCCFN), the contribution of MOA to the promotion of fruits and vegetable consumption was not adequate. More efforts and collaboration between MOA and MOH are needed to promote and enhance fruits and vegetable intake. 

About half of the world’s population eats rice as a staple meal [25]. Several studies have proposed that white rice intake leads significantly to the development of type 2 diabetes [26,27,28]; moreover, the consumption of germinated brown rice has been shown to postpone or inhibit the progression of obesity and chronic glucose resistance to type 2 diabetes [29,30,31]. Previous research proposed that the anti-obesity effects of germinated brown rice could be regulated in part by lipogenic gene downregulation [32]. A study also concluded that selecting a high-amylose cultivar and eating germinated rice reduces one’s predisposition to insulin resistance and obesity more than eating rice with only one of these properties for those who eat rice as a staple [33]. Hence, stakeholders should consider promoting germinated rice for a healthy diet to combat obesity in the country.

A policy, guidelines and an action plan on the marketing of food high in fat, sugar and salt to reduce the marketing to children of unhealthy foods (high in saturated fats, salt or sugar) were implemented and monitored. However, this policy is not widely communicated, and was unclear in terms of nutrition profiling and the criteria used to categorize “unhealthy” foods. Stakeholders expressed uncertainty in the feedback on the successes and challenges of implementing the voluntary compliance scheme and were unsure whether there had been a thorough review. More stringent rules should be enforced on the advertisement of foods with high fat, sugar and salt in the media [18]. 

### 4.3. Surveillance and Research

There is a national system for measuring physical activity. The Institute for Public Health, Malaysia collects information on physical activity via large population studies, albeit relying on crude self-reported data [6,7,8,17,19,21]. Better estimates of physical activity among different population groups are needed. There is a robust system in place to monitor physical activity in continuity. However, the quality of the measures used remains questionable. NHMS uses the short International Physical Activity Questionnaire that has its limitations.

In Malaysia, there is a national research program for preventing and controlling NCDs. Research priorities are set in Malaysia Plans, and health research is funded through three government agencies: the Ministry of Science and Technology Innovation (MOSTI), the Ministry of Higher Education (MOHE), and the Ministry of Health (MOH). The MOH funds applied and basic research through its lead health research agency, the National Institute of Health (NIH). The NIH comprises a network of research institutes. The MOSTI and MOE also fund applied and basic research. In addition to that, there should be dedicated funding toward furthering research in childhood obesity. Listing it as a research priority does not mean that these projects will be implemented. These lists are “wish lists” by various researchers. They serve as a reference to funding organizations, and some might not represent national research programs for preventing and controlling NCDs. What is needed is a more coordinated response among major stakeholders. 

### 4.4. Way Forward

With the current COVID-19 pandemic, it is crucial to look at implementing the policies related to childhood obesity, especially with low progress for all domains in the current study. A smartphone application for NCD education and risk evaluation can extend the preventive strategy scope in low- and middle-income countries [22]. However, it is impossible to anticipate the impact of direct-to-consumer health promotion and risk management where chronic illness treatment itself is uncertain and not uniform, and in the continuum where health care provider supervision can be a risk factor for illness. Will innovations be introduced? How will an approach which emphasizes wellness turn out? In a digital world crowded with promotion, how can promotion compete with tobacco, drink, and toxic commercials endorsing processed foods? The answers to these questions are likely to promise mobile health technologies relatively soon [3,23]. Several other opportunities for improvement include the utilization of “best buys” or other cost-effective interventions, a more sustainable NCD risk factor surveillance and finally, the need for additional funding for NCDs, particularly in research.

### 4.5. Strengths and Limitations

In the ASEAN region, including Malaysia, the prevalence of childhood overweight and obesity has risen dramatically. This paper shows that a country’s surveillance infrastructure needs prompt and reliable oversight, participatory analysis and assessment, and successful solutions, both of which must be tackled in order to leverage the country’s whole-of-government and whole-of-society approach to tackling childhood obesity. 

To the best of our knowledge, this is the first study that describes the NCD policy interventions and measures performance across three areas—governance, risk factors and surveillance/research among various stakeholders from different ministries, organizations, academia, and industries on the implementations of childhood obesity-related policies in Malaysia. Our findings serve as an important reference, especially to the policymakers, to review all the related policies implemented in the country. This study also suggests that policies should be reviewed promptly to ensure the target populations’ effectiveness.

The Delphi method, which has the benefit of anonymity and a knowledge feedback loop and uses statistical inference, is widely used. The Delphi survey was found to be an effective tool for leaders, experts and relevant stakeholders. This study is, however, not without its limitations. Using the Delphi approach, we encountered difficulties with the nature of questionnaires, the evaluation of experts, the bias of participants, and the lack of representation [34]. Another bias in this study may be the uneven percentages or various subjective understandings of stakeholders regarding the childhood obesity policies.

## 5. Conclusions

Our study found that most of the policies on childhood obesity implementation experienced low progress. Governmental bodies, academia, organizations and agencies in Malaysia have acknowledged that tackling NCDs is a national priority for social development. Malaysia needs to implement longer-term strategies that are affordable, practical and consistent with NCD prevention and control. An irresistible overall pattern is indeed the involvement of appropriate government departments, organizations, health practitioners, civil society and the private sector. While Malaysia has released several key strategic documents on childhood obesity and introduced some policy initiatives, we have identified some gaps that need to be addressed to maximize the whole-of-government, whole-of-society approach to combat childhood obesity in Malaysia.

## Figures and Tables

**Figure 1 ijerph-18-05950-f001:**
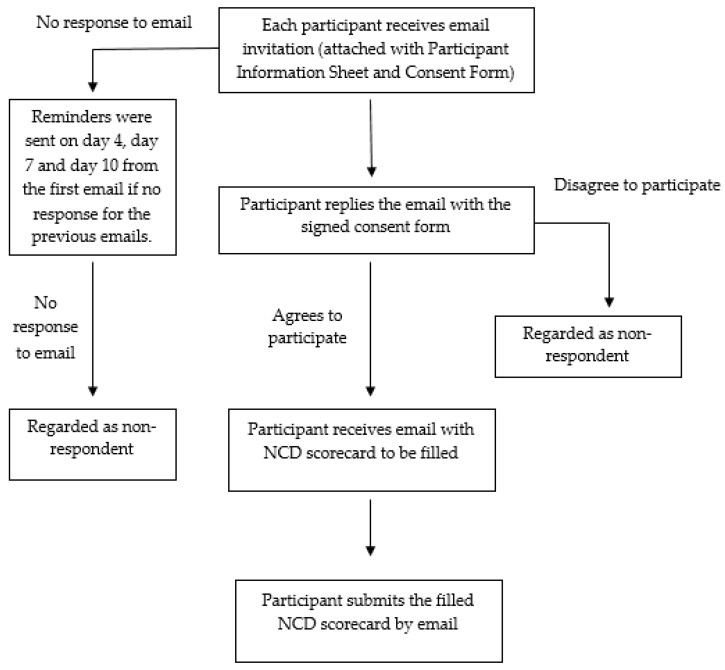
Data collection flow chart.

**Table 1 ijerph-18-05950-t001:** Classification of the NCD scorecard progress by domain.

Score	Interpretation of Progress
20% or less	Very low progress
21–40%	Low progress
41–60%	Moderate progress
61–80%	High progress
More than 80%	Very high progress

**Table 2 ijerph-18-05950-t002:** Percentage of progress for each indicator in the three domains.

Domain	Number of Indicators	% of Progress
Governance	9	23.0
Risk factors	4	23.5
Surveillance and research	2	25.6
Health system response	-	-

**Table 3 ijerph-18-05950-t003:** Number of respondents and percentage of progress in the various domains and its indicators.

Domain	Item	Indicators	Policies/Programs/Plans	% of Progress	*n* (%)
Governance	1	Inclusion of NCD in the national health plan and/or your national development agenda.	Policy Options to Combat Obesity in Malaysia 2016–2025	2.51	19 (86.4)
			b. National Strategic Plan for Active Living 2016–2025	2.78	21 (95.5)
	2	Existence of unit/branch/department in the Ministry of Health or equivalent with NCD responsibility that is well-resourced.	Disease Control Division	2.51	19 (86.4)
			b. Family Health Development Division (BPKK)	2.51	19 (86.4)
			c. Health Education Division	2.78	21 (95.5)
			d. Nutrition Division	2.52	19 (86.4)
	3	There are national projects, programs and initiatives in the health sector to mobilize resources, build capacity, train health workers, and exchange information, lessons learnt and best practices on reducing the burden of NCD.	School Health Program	3.18	19 (86.4)
			b. Adolescent Health Program—including mental health component (inc. suicides)	3.04	18 (81.8)
			c. Prioritizing Food Policy Options to reduce Obesity in Malaysia 2015—ASM	3.04	19 (86.4)
Risk factors	4	There is a policy, strategy, or action plan to reduce the consumption of high content sugar beverages in school-aged children and adolescents that are implemented and regularly monitored.	National Plan of Action for Nutrition of Malaysia (NPANM) III 2016–2025—Nutrition Programs and Activities for School Children	6.84	21 (95.5)
	5	There is a policy, strategy, or action plan to increase the consumption of fruit and vegetables implemented and monitored.	National Plan of Action for Nutrition of Malaysia (NPANM) III 2016–2025	6.85	21 (95.5)
			b. National Agro-Food Policy, 2011–2020	6.25	17 (77.3)
	6	There is a policy, strategy, or action plan to reduce the marketing of unhealthy foods to children (high in saturated fats, salt, or sugar) implemented and monitored.	Guidelines on Marketing of Food High in Fat, Sugar and Salt (HFSS), Malaysia	6.55	20 (90.9)
Surveillance and research	7	There is a national system for measuring physical activity.	Institute of Public Health (IKU) Malaysia collects information on physical activity via large population studies, e.g., National Health Morbidity Surveys—1986, 1996, 2006, 2011, 2015, 2019.	13.09	21 (95.5)
	8	There is a national research program for preventing and controlling NCD.	Research priorities are set in Malaysia Plans, and health research is funded through three government departments—MOH, MOSTI and MOHE. The Ministry of Health funds applied and basic research through its lead health research agency, the National Health Institutes, which comprises a research institute network. The Ministry of Science, Technology and Innovation and the Ministry of Education also fund applied and basic research.	12.50	20 (90.9)

## Data Availability

The data presented in this study are available on request from the corresponding author. The data are not publicly available as the data are currently being analysed in other related papers.

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
