# Peer review of "The Implementation of Childhood Obesity Related Policy Interventions in Malaysia—A Non Communicable Diseases Scorecard Project"

_ijerph, 2021, doi:10.3390/ijerph18115950_

Round 1
Reviewer 1 Report
The prevalence of overweight and obesity among children has increased tremendously in the ASEAN region, including Malaysia. This paper reveals that timely and accurate monitoring, participatory review and evaluation, and effective remedies are necessary for a country's surveillance system, which must be addressed to leverage the whole-of-government and whole-of-society approach in addressing childhood obesity in the country. These results are of great theoretical significance and practical value for solving overweight and obesity among children. The analysis process is comprehensive, good organized, large amount of information and so on. Minor revision can be published in International Journal of Environmental Research and Public Health. Existing papers are greatly improved over the original manuscript ijerph-1122840-peer-review-v1. However, there are some minor issues need to be improved: Authors should revise the format of reference according to IJERPH Journal.

Reviewer 2 Report
In my opinion the manuscript entitled „The Implementation of Childhood Obesity Related Policy Interventions in Malaysia – A NCDs Scorecard Project” cannot be published because it does not make a significant contribution to the issue of obesity in Malaysian children.
I want to emphasize that obesity of children from Malaysia has been the subject of many studies, examples below:
- Koo et al. (2020) “The GReat-Child TrialTM: A Quasi-Experimental Dietary Intervention among Overweight and Obese Children”
- Poh et al. (2019) Low socioeconomic status and severe obesity are linked to poor cognitive performance in Malaysian children
- Mahaletchumy et al. (2019) “Prevalence of overweight/obesity and its associated factors among secondary school students in semi urban area in Malaysia”
- Yusop et al. (2018) “The effectiveness of a stage-based lifestyle modification intervention for obese children”
- Norimah et al. (2014) “Association of Body Weight Status and Socio-Demographic Factors with Food Habits among Preschool Children in Peninsular Malaysia”
- Firouzi et al. (2014) “Sleep habits, food intake, and physical activity levels in normal and overweight and obese Malaysian children”
- Naidu et al. (2013) “Overweight among primary school-age children in Malaysia”
- Hamzaid et al. (2011) “Quality of life of obese children in Malaysia”
- Ong et al. (2010) “Factors associated with poor academic achievement among urban primary school children in Malaysia”
- Sidik & Ahmad et al. (2004) “Childhood Obesity: contributing factors, consequences and intervention”
Authors provided data obtained only from 22 participants. It is not known how the participants were selected, it is not known for whom the sample is representative and finally it is not known why their opinions are to be considered as contributing something significant. It seems that it would be more reasonable to prepare a meta-analysis of data from available scientific studies on obesity in children and adolescents in Malaysia.
Author Response
Please see the attachment. Thank you!

This manuscript is a resubmission of an earlier submission. The following is a list of the peer review reports and author responses from that submission.
Round 1
Reviewer 1 Report
Dear author:
Here are a number of comments that you should include in your paper:
It is not advisable to include abbreviations in an article before putting them in full. You have done this directly in the title with "NDCs".
In your research you include subjective aspects that complement the DELPHI technique. He talks about a complementary interview, but does not explain how it contributes to the conclusions of the study. Nor does it indicate the profile of the 22 experts for the use of the DELHPI technique and the answers given in the two rounds stipulated by the method. The following papers are recommended reading:
Vio Fernando, Lera Lydia, Fuentes-García Alejandra, Salinas Judith. Método Delphi para identificar materiales educativos sobre alimentación saludable para educadores, escolares y sus padres. ALAN [Internet]. 2012 Sep [citado 2021 Feb 21] ; 62( 3 ): 275-282. Avialable: http://ve.scielo.org/scielo.php?script=sci_arttext&pid=S0004-06222012000300010&lng=es.
Cruz Ramírez Miguel. Un estudio sobre la implementación del método Delphi en publicaciones de ciencias médicas indexadas en Scopus. Educ Med Super [Internet]. 2018 Sep [citado 2021 Feb 21] ; 32( 3 ): 36-50. Avialable: http://scielo.sld.cu/scielo.php?script=sci_arttext&pid=S0864-21412018000300003&lng=es.
Regarding the paper, the section devoted to the explanation of the study design (2.2) is poorly developed, summarising it in a single sentence.
Sections 2.3, 2.4 and 2.5 are part of the DELPHI methodology and do not add anything new to the science. They could have been briefly described and represented as a table or flow chart. The experts' response to the second survey is not included.
Section 3 on results is devoted more to the response of the respondents than to the innovative result of the work itself.
In the discussion section, the authors have focused them appropriately to develop a scorecard with indicators of the key factors. The problem with the work is that the procedure followed has not developed the double survey that is required by the DELPHI method and therefore lacks robustness.
Finally, the conclusions are obvious and make no special mention of the work carried out or of future lines of action to promote the reduction of childhood obesity in Malaysia.
Best regards
Reviewer 2 Report
The prevalence of overweight and obesity among children has increased tremendously in the ASEAN region, including Malaysia. This paper reveals that timely and accurate monitoring, participatory review and evaluation, and effective remedies are necessary for a country's surveillance system, which must be addressed to leverage the whole-of-government and whole-of-society approach in addressing childhood obesity in the country. These results are of great theoretical significance and practical value for solving overweight and obesity among children. The analysis process is comprehensive, good organized, large amount of information and so on. Minor revision can be published in International Journal of Environmental Research and Public Health. However, there are some major issues need to be improved:
- Abstract:Lack of the most effective techniques to address overweight and obesity in children;
- Introduction: Updated reference to supplement refined staple foods leading to obesity,such as https://www.mdpi.com/1422-0067/19/9/2785;https://www.hindawi.com/journals/omcl/2020/3836172/
- Discussion: Obesity of low GI brown rice into high GI refined rice has not attracted the attention of Asia to solve the obesity problem, especially rice as the staple food to ignore low GI grains (such as barley) in This factor should be added to the discussion to provide decision-making advice for the Malaysian government and society.
- References: Authors shouldrevise the format of reference according to IJERPH Journal.
Reviewer 3 Report
In my opinion the manuscript entitled „The Implementation of Childhood Obesity Related Policy Interventions in Malaysia – A NCDs Scorecard Project” cannot be published because it does not make a significant contribution to the issue of obesity in Malaysian children.
I want to emphasize that obesity of children from Malaysia has been the subject of many studies, examples below:
- Koo et al. (2020) “The GReat-Child TrialTM: A Quasi-Experimental Dietary Intervention among Overweight and Obese Children”
- Poh et al. (2019) Low socioeconomic status and severe obesity are linked to poor cognitive performance in Malaysian children
- Mahaletchumy et al. (2019) “Prevalence of overweight/obesity and its associated factors among secondary school students in semi urban area in Malaysia”
- Yusop et al. (2018) “The effectiveness of a stage-based lifestyle modification intervention for obese children”
- Norimah et al. (2014) “Association of Body Weight Status and Socio-Demographic Factors with Food Habits among Preschool Children in Peninsular Malaysia”
- Firouzi et al. (2014) “Sleep habits, food intake, and physical activity levels in normal and overweight and obese Malaysian children”
- Naidu et al. (2013) “Overweight among primary school-age children in Malaysia”
- Hamzaid et al. (2011) “Quality of life of obese children in Malaysia”
- Ong et al. (2010) “Factors associated with poor academic achievement among urban primary school children in Malaysia”
- Sidik & Ahmad et al. (2004) “Childhood Obesity: contributing factors, consequences and intervention”
Authors provided data obtained only from 22 participants. It is not known how the participants were selected, it is not known for whom the sample is representative and finally it is not known why their opinions are to be considered as contributing something significant. It seems that it would be more reasonable to prepare a meta-analysis of data from available scientific studies on obesity in children and adolescents in Malaysia.